# Factors Influencing the Training Process of Paralympic Women Athletes

**DOI:** 10.3390/sports11030057

**Published:** 2023-02-28

**Authors:** Manuel Rodríguez Macías, Francisco Javier Giménez Fuentes-Guerra, Manuel Tomás Abad Robles

**Affiliations:** Facultad Educación, Psicología y Ciencias del Deporte, Av. Alcalde Federico Molina Orta, 21007 Huelva, Spain

**Keywords:** parasport, sport training, woman athlete, paralympic games, barriers and facilitators

## Abstract

(1) Background: Paralympic women athletes in their training process go through a series of interrelated stages which are parallel to their evolutionary development, during which a wide variety of psychological, social, and biological factors will have an impact. The purpose of this study was to analyze the factors influencing the sports training process of Spanish Paralympic women athletes (social, sporting, psychological, technical–tactical factors, physical condition, as well as barriers and facilitators) who had won at least one medal (gold, silver, or bronze) in the 21st century Paralympic Games (from Sydney 2000 to Tokyo 2020). (2) Methods: The research involved 28 Spanish Paralympic women athletes who had won at least one medal at a Paralympic Games in the 21st century. An interview of 54 questions grouped into 6 dimensions (sport context, social context, psychological, technical–tactical, physical fitness, and barriers and facilitators) was used. (3) Results: Coaches, as well as families, were essential in the sport development of Paralympic athletes. In addition, most women athletes recognized that psychological aspects were of vital importance, as well as working on technical–tactical aspects and physical fitness in an integrated way. Finally, the Paralympic women athletes highlighted that they had to face numerous barriers, mainly financial challenges and issues with visibility in the media. (4) Conclusions: Athletes consider it necessary to work with specialists to control emotions, increase motivation and self-confidence, as well as to reduce stress and anxiety and manage pressure. Finally, the training process and sporting performance of Paralympic women athletes are conditioned by several barriers, including economic, social, architectural, and disability barriers. These considerations can be taken into account by the technical teams working with Paralympic women athletes, as well as by the competent bodies, to improve the sports training process of these athletes.

## 1. Introduction

### 1.1. The Process of Sports Training

The training process of women athletes goes through a series of stages, parallel to its evolutionary development [1]. This process is influenced by a wide variety of psychological, social, and biological factors [2]. Giménez [3] affirms that the athlete will go through a series of stages in which she will learn and be trained in different contents (physical, technical, tactical, and psychological) adapted to her biological and psychological characteristics. Cárdenas and López [4] consider that sports training is influenced by methodological, psychological, genetic, and physiological factors, which significantly affect the training of athletes. Additionally, Castejón et al. [5] consider that, in addition, the motor, cognitive, and affective aspects of each athlete should be taken into account. In this sense, Lorenzo and Calleja [6] suggest analyzing the process of training high-performance athletes and the factors that influence it, to find the optimal conditions for the development of competencies that can be applied to the process of training newly talented athletes. In this way, we could ask about the main factors that can influence the training process of Paralympic athletes.

### 1.2. Factors Influencing the Training Process of Paralympic Women Athletes

When talking about the different factors influencing the training process of Paralympic athletes, different aspects that positively or negatively influence their training process are often highlighted. Ruíz and Salinero [7] state that, nowadays, athletes and their performance cannot be understood without an integral conception of themselves and their environment, which is why their sporting success can hardly be perceived as a purely personal achievement. The influence of sport and social contexts, psychological, technical–tactical, and physical fitness aspects in the training process of athletes has been discussed in the scientific literature [8]. In this regard, Willis et al. [9] highlight the relevance of the social contexts, Martinent and Decret [10] emphasize aspects related to psychology and fitness, while Rimmer et al. [11] consider that there are many physical, emotional, and psychological barriers influencing the training and sports development of athletes with disabilities. Even though there are studies on the training process of athletes [12,13,14,15], there is a lack of research analyzing the sports training process of female and male Paralympic athletes, as well as the influence of the different factors throughout their sporting life from the perspective of the athletes themselves.

### 1.3. Need for Research

Studies on the triangle of gender, disability, and sport have little presence in the specialized literature [16] despite the great growth of the Paralympic Games and the incorporation of women in this event [17]. Athletes with disabilities have to overcome structural, social, and economic barriers. In addition, Willis et al. [9] highlight the existence of a very limited understanding of the mechanisms and processes of training disabled athletes that allow them to participate in competitive-oriented physical activities. Knowledge of all these aspects can help coaches, psychologists, and other professionals to improve the sports training of these athletes [8,18]. The scarcity of research focused on the analysis of these factors, and in this context, justifies this work. Additionally, this study is justified because it can contribute to increasing the visibility of women athletes with disabilities, not only among researchers, academics, and sports training professionals, but also in the rest of society, promoting awareness of the importance of the role of these women athletes. The qualitative approach was selected to examine the way in which the subjects perceived and experienced the phenomena that surrounded them, deepening in their views, interpretations, and meanings, and because the subject of the study has been little explored [19]. The main question raised in this research is the following: what are the factors that influence the training process of Spanish Paralympic women athletes? In this sense, this research aimed to analyze the factors influencing the sports training process of Spanish Paralympic women athletes (social, sporting, psychological, technical–tactical factors, physical condition, as well as barriers and facilitators) who had won at least one medal (gold, silver, or bronze) in the 21st century Paralympic Games (from Sydney 2000 to Tokyo 2020).

## 2. Materials and Methods

### 2.1. Participants

The research involved 28 Spanish Paralympic women athletes (mean age 38.57 ± 8.15) who had won at least one medal (gold, silver, or bronze) at a Paralympic Games in the 21st century (Sydney 2000, Athens 2004, Beijing 2008, London 2012, Rio de Janeiro 2016, and Tokyo 2020). The total number of medals won by the women interviewed was 65, including 18 gold medals, 23 silver medals, and 24 bronze medals. As for the sports practiced, these were: swimming, athletics, judo, goalball, and cycling. The athletes had visual, intellectual, or physical disabilities. It should be noted that the same athlete may have participated in more than one Paralympic Games.

### 2.2. Instrument

The data were collected using a qualitative research methodology [20], with the aim of obtaining a detailed description of the research context, maximum objectivity, and correct data collection according to the objective set out in the study. The research instrument used was the interview designed and validated by Robles et al. [21]. This tool was used because the researchers knew it well and hoped, as in the past, to obtain robust results after its application. With the idea of completing this tool, the dimension “facilitators and barriers” was added, due to the importance of the subject for athletes with disabilities and due to its scarce study in the specialized literature. It should be noted that some of the advantages of using an interview are that the interviewees can provide historical information and, in addition, it allows some control of the interviewer when it comes to including and addressing the issues and questions. Regarding the limitations of the use of interviews, it highlights the fact that the data are “filtered” by the views of the participants and that not all subjects have the same abilities to express themselves verbally [17]. The interview consisted of 54 questions grouped into 6 dimensions: -Sport context. This dimension sought to determine the age at which the interviewees started practicing sport, their memories (positive or negative), practice of other sports, considerations related to the different clubs in which they had trained, reasons why they started practicing sport or competing, aspects related to coaches, and considerations related to competition and sporting performance.-Social context. The aim of this dimension was to understand the perceptions of Paralympic women athletes about their close environment (family), studies and/or academic training, friends and/or teammates, and finally, the media. -Psychological aspects. The aim was to analyze whether the women athletes have worked with psychologists and, if so, how. In addition, the objective was to find out which psychological aspects were most important to them: emotions, motivation, self-confidence, pressure, stress, and anxiety. Emotions are understood as a complex reaction pattern, which involves experiential, behavioral, and physiological elements; motivation is defined as the impulse that gives purpose to behavior; self-confidence refers to confidence in one’s own abilities; pressure is understood as excessive or stressful demands that influence the way one thinks, feels, or acts; stress is the physiological or psychological response to internal or external distressing factors; and anxiety is characterized by the learning of emotions and behaviors and the somatic symptoms of tension that a person may manifest [22].-Technical–tactical aspects. The purpose was to determine the importance given to these aspects by the women athletes, whether they considered that they have worked on them to the right extent during their sports training, and at what stage they considered it necessary to further deepen their knowledge of them.-Physical fitness. The goal was to study how they developed their physical fitness during their sports training, who was responsible for their preparation, what physical capacities they considered most important, and how they thought they should be trained during the sports training process.-Facilitators and barriers. This was aimed at analyzing the opinion of women athletes on the factors which may influence the training process, as well as the aspects which should be taken into consideration in an ideal training model. Furthermore, it sought to detect the different difficulties they had encountered in their sporting careers, both because they were women and because they had a disability, as well as other barriers they may identify. This research will focus on the most important aspects of each of the dimensions. Table 1, Table 2, Table 3, Table 4, Table 5 and Table 6 show the categories, descriptions, and examples of the dimensions of the interview.

### 2.3. Procedure

Firstly, the Spanish Paralympic Committee was contacted to obtain, confidentially, the data of the Paralympic women athletes who won at least one medal (gold, silver, or bronze) in some of the Paralympic Games from Sydney 2000 to Tokyo 2020. An initial contact was then made via email, where a letter was sent to each athlete informing them of the objectives and reasons for the study and requesting their participation in the study. Once acceptance was received from each of the Paralympic women athletes, contact was made by telephone to arrange a date and time for the interview. Finally, another telephone call was conducted with each of them to carry out the interview. The interviews were performed by telephone due to the difficulty of accessing each of the women athletes personally, geographically located across Spain, and also due to the health situation (the COVID-19 pandemic). The interviewees were informed of the purpose of the study and were guaranteed the confidentiality of the information collected. The study is framed within the international ethical declarations of Helsinki (2013), the recommendations of the World Health Organization, the code of ethics, the regulations on data confidentiality, and the Organic Law 3/2018 of 5 December on Data Protection and guarantee of digital rights. The research was conducted with the consent of the Research Bioethics Committee (PEIBA, 1915-N-19) of Andalusia (Spain).

### 2.4. Statistical Analysis

The interviews were taped with a computer recorder (MacBook Air 2020) and transcribed verbatim into a word processor (Microsoft Word for Mac 16.50). The interviews were analyzed with the specialized qualitative research software MAXQDA (20.4.0). In order to make the analysis more reliable, coding was carried out by a group of 3 coders, which was made up of experts in the field of interview coding and qualitative research. Following the guidelines proposed by McPhail et al. [23] regarding the codification process in qualitative research, internal inter-rater reliability was estimated by means of interobserver agreement (IOA) [24], reaching optimal interobserver reliability (91.01%) [25] in the third coding and analysis meeting, which allowed group members to code the interviews individually. Additionally, to reduce the agreements resulting from chance alone, the Kappa Index [26] was calculated using SPSS 27.0 software. The value was 0.889 with *p* < 0.05 at the third coding and analysis meeting, which is considered almost perfect [27]. Finally, to obtain greater validity, all the research interviews were divided among the members of the coding group, so that each interview was analyzed and coded by two people.

## 3. Results

### 3.1. Sport Context Dimension

As for the age when they started practicing sport, the mean age was 12.11 years (SD = 7.10). On the other hand, 50% (*n* = 14) of the women athletes mainly remembered how much fun it was to play the games during training when they were in their formative years.

I remember the training sessions were fun. There were a lot of us in the club and I had a lot of fun with my teammates. I also remember that we played a lot of games, especially at the beginning of the season and at the end of the season(woman athlete 13, paragraph 6)

92.86% (*n* = 26) of the interviewees had played other sports, especially in physical education lessons, and 61.54% (*n* = 16) felt that this had a positive influence on them.

This has had a positive influence on me. It has helped me in my training as an athlete(woman athlete 3, paragraph 3)

Concerning their stay in clubs throughout their sporting career, 10 interviewees (35.20%) stated that they had always stayed in the same club, while the remaining 64.80% (*n* = 18) had been in several clubs.

Well, I remember a lot of club changes until I found a very good coach at a club that allowed me to qualify for my first Paralympic Games(woman athlete 10, paragraph 4)

On the other hand, 100% (*n* = 28) of the Paralympic women athletes highlighted the importance of the figure of the coach in their training process and also that the coach had adequate training.

My last two coaches have also been very important, both the one at my current club and the one at the high-performance center, with whom I have a very good relationship. They have been vital for me. Additionally, they still are, to this day(woman athlete 5, paragraph 12)

I changed coaches several times. I changed the first one because at a certain point situation was too much for him. He was an amateur coach. He did not feel he was capable of coaching me anymore(woman athlete 4, paragraph 10)

Regarding the age at which they started to participate in high-level competition, 71.43% (*n* = 20) of the athletes started at puberty (between 8 and 14 years). Furthermore, the length of time the women athletes had been competing at a high level ranged from 4 to 20 years. On the other hand, the athletes underlined the importance of being motivated to compete and the fact of competing to be with friends and to win medals.

My level of motivation was the same as during training. I was doing the same as I had trained. I did not arrive too activated because of what I mentioned before, it could backfire on me. You could say that my motivation was the same in training as in competition(woman athlete 7, paragraph 30)

Additionally, the teammates, above all, because my motivation for going to compete was not the fact of competing, but meeting friends(woman athlete 1, paragraph 12)

Seeing myself with the possibility of winning medals at the international level. The following year was the Paralympic Games in Rio de Janeiro and it was an opportunity I did not want to miss(woman athlete 5, paragraph 8)

### 3.2. Social Context Dimension

For 100% (*n* = 28) of the Paralympic athletes, the family was essential throughout their training process as athletes, as well as the entire social context surrounding them.

It is crucial. To reach high performance you need to feel supported and valued by the people around you. Moreover, even though you are doing something for yourself and that you feel good doing it, at the end of the day, an athlete is always looking for recognition. It is important to notice this recognition for results, effort, dedication... It is essential that the social context supports you. Additionally, then I think that, at the Paralympic level, the social and media context is beneficial for everyone(woman athlete 13, paragraph 20)

25% (*n* = 7) highlighted the influence exercised by their family members who had played or played sports.

Yes, they have played sports. My brother is a swimmer and a water polo player, so we are in the same boat. Additionally, my parents also played a sport, but as amateurs(woman athlete 17, paragraph 21)

Very much so. I was very familiar with swimming. Additionally, as I mentioned before, the fact that my sister was a swimmer means that she understands you better, that she knows what you are talking about, she supports you, helps you, and walks with you in your suffering, in your travels and in everything. It was very good for me because she knew how to help me better(woman athlete 13, paragraph 21)

64.28% (*n* = 18) of the athletes interviewed said they had a university education. In terms of relevance, 11 women athletes interviewed (39.28%) considered their studies to be more important than their sport. 

They have always been above sport. There have been times when I have valued sport more, especially when I was already at university when I had a bit more freedom and it was me who was paying for my studies. I told them not to worry because I was the one who was paying for my studies and that I preferred to study at my own pace so that I could combine it with the sport. I would decide whether I wanted to take more or fewer subjects(woman athlete 11, paragraph 14)

Sport came before my studies. There was a moment when I decided that I could either get my degree or fight for a medal at the Games, that I had chance to fight for it(woman athlete 21, paragraph 14)

It was also interesting to observe the opinion of the interviewees concerning combining studies and sport. In this sense, 100% (*n* = 28) stated that it was important to balance studies and sports.

It’s very important to balance them because if you do not study, what do you do for a living? In fact, if you do not study, when you finish your sporting career, you will find yourself with nothing. Sport has an expiration date. It is essential to combine both things. High-level sport comes to an end and if you do not have studies, you are left with nothing; no job, no studies...(woman athlete 25, paragraph 15)

The women athletes underlined the importance of their friends and fellow athletes, and all 28 women athletes (100%) stressed that it was important to reconcile friendships with the sport.

I think it is important. After so many years in the sport, I have had moments when you are worse and less motivated or you have personal and social problems, and at the end that ends up affecting your performance, training, and competition(woman athlete 21, paragraph 20)

For me, it was easy to combine because I had friends in the club and on the national team. I also had friends outside. Depending on the time of the season regarding the competition, I could socialise more or less, but I already had social relations in the same environment(woman athlete 18, paragraph 17)

It’s not easy to balance them. You can combine them by making friends who do the same thing, even if they do not do the same sport, but who have the same discipline as you, and the same lifestyle, because otherwise, it’s complicated(woman athlete 4, paragraph 17)

For some of the athletes interviewed (17.86%; *n* = 5), the media were largely responsible for the lack of adequate visibility of Paralympic sports, let alone women’s Paralympic sports.

Let us start with the visibility of both Paralympic and women’s sports. Have you been able to watch the Paralympics on TV? I go crazy looking for the channel, and where is women’s football? I know there are economic interests, but if girls do not watch sports, how are they going to have references and how are they going to practice sport? It is neither known nor sold(woman athlete 12, paragraph 52)

You can cry over this double formula. If you are a Paralympic athlete and on top of that you are a woman, you do not exist, you do not have any visibility(woman athlete 25, paragraph 51)

### 3.3. Psychological Dimension

100% (*n* = 28) of the Paralympic women athletes stated that the importance of psychological aspects was high, not only during the training process but also when they reached the elite level, emphasizing motivation, perseverance, and discipline, among others.

All the importance. I think you have to be very strong in your mind. In my opinion, when it comes to competing you are 70% mental and 30% physical. I know people who train very little and then compete brilliantly because of their mental strength and people who train a lot and when it comes to competing they are blocked and do not get good results in competition. I see the psychological aspects as essential(woman athlete 5, paragraph 22)

Motivation. Motivation is key(woman athlete 1, paragraph 23)

Well, I would highlight determination, perseverance, and being a mentally strong person. I am also, a disciplined and optimistic person, who always approached competitions with confidence and optimism(woman athlete 21, paragraph 23)

Furthermore, 53.57% (*n* = 15) of the women athletes used to work with sports psychologists, while 3.57% (*n* = 1) worked on psychological aspects with their coaches or trainers. Thus, for all 28 women athletes (100%), it was important to work with sport psychologists.

Many teammates work with psychologists, but it is not something that is forced on us. We can ask for it whenever we want at no cost(woman athlete 3, paragraph 24)

I do not know, I do not even remember the name. My psychological work has been more with coaches than with psychologists(woman athlete 21, paragraph 26)

Very important. The fact that I can rely on a psychologist helps me to manage situations not only on a sporting level, but also a personal level(woman athlete 5, paragraph 24)

60.71% (*n* = 17) of the women athletes revealed that both the days before and the day of the competition, they felt nervous and that they could not control their emotions.

Very bad. Insomnia, anxiety, fatigue, general discomfort...(woman athlete 4, paragraph 28)

I was much more nervous and stressed that day(woman athlete 23, paragraph 28)

Yes, these negative emotions usually come to me when I have done a test wrong or made a big mistake. I try to control it, although you do not always succeed 100% because it overwhelms you(woman athlete 3, paragraph 31)

I did not use to or do not usually control my emotions and that is why I was going through what I was going through(woman athlete 5, paragraph 31)

On the other hand, 71.43% (*n* = 20) of the interviewees highlighted the great importance of motivation and self-confidence, especially in competitions.

During the championships, my motivation level is even higher than during training because it is the culmination of everything you have worked on. Do you really want or train for those moments(woman athlete 3, paragraph 30)

Quite high. In general, I was self-confident, except for when I had the injury, which I did find more difficult. However, as you go on competing, you convince yourself that you are fine(woman athlete 11, paragraph 32)

In terms of pressure, stress, and anxiety, 46.43% (*n* = 13) of the women athletes reported that they perceived pressure from the coach, while 17.86% (*n* = 5) had self-imposed pressure.

Yes, I felt pressure coming from the coaches. I felt pressured, but for me, it is not a bad thing(woman athlete 4, paragraph 19)

I put pressure on myself. It’s complicated. For example, if I get a gold medal, they give me a scholarship and in every competition you have to get a gold medal otherwise they take it away. That money helps with expenses(woman athlete 9, paragraph 19)

### 3.4. Technical–Tactical Dimension

For all the Paralympic women athletes (100%), both technique and tactics were important in their training as athletes, and they considered that these aspects should be worked on in the initial stages of sports training.

It is essential. I learned the technique wrong and after 5 years as a judoka I had to redo my entire technical background. It was a very complicated process, but I was able to overcome it with flying colors before I reached my first Games(woman athlete 7, paragraph 36)

Just like the technique. It is very important, and we prepared the matches depending on the opponent we were playing against. Certainly, it is as important as the technique(woman athlete 14, paragraph 39)

The idea is in the initial stages because that is when you can best absorb the learning. There are things that I did not get to learn well because I already had vices and I’ve been doing them badly all my life. The idea is to start in the early stages, although you can always improve(woman athlete 13, paragraph 37)

### 3.5. Physical Fitness Dimension

As for the role of physical fitness in the sports training process of the Paralympic women athletes, 100% (*n* = 28) of the interviewees considered this to be crucial. A total of 14.29% (*n* = 4) stated that they had worked with physical trainers, while the remaining 85.71% (*n* = 24) acknowledged that it was their coach who was in charge of carrying it out. In addition, seven of them (25%) claimed to have worked on their physical fitness in an independent way, not integrated with the other training contents, while the remaining 75% (n = 21) asserted that they had carried it out in an integrated way.

It is fundamental, it is the basis of everything. If you do not have good physical preparation, you have nothing to do(woman athlete 25, paragraph 43)

We had a physical trainer in addition to the coach(woman athlete 19, paragraph 43)

The coaches I have had at any given time. We did not have a specific physical trainer(woman athlete 11, paragraph 43)

We did specific sessions to train physical fitness. We worked on endurance with continuous running. Additionally, then we worked on strength in the gym with the routines we received(woman athlete 20, paragraph 47)

### 3.6. Facilitators and Barriers

100% (*n* = 28) of the Paralympic women athletes considered their sporting success to be based on different aspects, among which the following stood out: discipline, perseverance, training, capacity for effort, and support from family, coaches, teammates, and friends.

The training, the discipline, the physical and emotional effort, the unconditional support of the people around you and the friendships in the team. That is why I have stayed for so many years(woman athlete 1, paragraph 49)

The trust placed in me by the coaches and then my determination, perseverance and dedication(woman athlete 6, paragraph 49)

When the interviewees were asked if they would make changes in their training process, 67.86% (*n* = 19) answered affirmatively. In addition, it is worth noting that some of them proposed that they should have more fun doing sport in the training process, that they would take away the pressure to achieve results too soon, and they would work more with sports psychologists.

Well, I’ve thought about it a thousand times. Psychologically I would try to enjoy it more, I would try not to be so competitive. It would have been different to have had 10 years in high-level performance that was more relaxed and more enjoyable and not focused on the result and the objective of winning, with all the tension that entails. I think the medal does not make up for it(woman athlete 14, paragraph 53)

On the other hand, 67.86% (*n* = 19) of Paralympic women athletes acknowledged that they had more barriers due to disability than gender. In addition, they felt that they had to adapt to move forward, as they were difficult to overcome.

I think it was more because of my disability than because I am a woman, but there are certainly barriers for women in sports. In fact, I think there are many differences between men’s and women’s sports, especially on a commercial level, in terms of sponsorship and prizes. The remuneration is not the same either. In the end, men’s sport moves more money. Additionally, if you move more money, you get paid more. If women’s sport was promoted, could it move more money? Sure(woman athlete 16, paragraph 51)

You never get over them because they are always there. You have to settle, in inverted commas, and if you want to get to the top, you have to train as best you can and make a living. You cannot get down because of that, you have to pull yourself up by your bootstraps(woman athlete 2, paragraph 51)

Finally, 53.57% (*n* = 15) of the interviewed women referred to the economic barriers Paralympic women athletes have to endure, as well as the scarce visibility and social recognition of their successes.

On an economic level, there are many differences between Paralympic and Olympic sports, we do not have the same social recognition, nor the same recognition in the media(woman athlete 11, paragraph 51)

We also do not receive the same scholarships and recognition as other non-disabled athletes(woman athlete 9, paragraph 51)

Figure 1 shows a summary of the main factors influencing the training process of Paralympic women athletes.

## 4. Discussion

This research aimed to analyze the factors influencing the sports training process of Spanish Paralympic women athletes (social, sporting, psychological, technical–tactical factors, physical condition, as well as barriers and facilitators) who had won at least one medal (gold, silver, or bronze) in the 21st century Paralympic Games (from Sydney 2000 to Tokyo 2020).

Regarding the sporting context dimension, the age at which the women athletes interviewed started practicing sport varied, with the average age being close to puberty. However, according to Ruíz [28], it is from the age of six when sports practice should begin. On the other hand, many of the women athletes interviewed had gone through several clubs during their sporting careers, which may be due to a lack of training planning on part of the trainers or coaches [29]. Furthermore, the figure of the coach is essential throughout their careers, influencing both their training process and sporting performance [30,31,32]. 

As regards the social dimension, the family was fundamental in the training process and in the women athletes’ performance, as shown in the studies by Nuviala et al. [33], Reinboth and Duda [34], Echazarreta et al. [35], Sánchez-Miguel et al. [36], and Robles et al. [14]. Some of the women athletes interviewed considered the fact that their family members played sports to have influenced them to practice sports themselves. This aspect is highlighted by Carrasco [37], who states that the way young people behave is strongly related to the actions of their families. However, families can also exert pressure and adopt inappropriate behaviors which may lead to the abandonment of sport [38]. However, most of the athletes stated that they were not influenced by family members when starting in the sport, which seems to indicate that this factor does not seem to be decisive for becoming to a Paralympic medalist. Additionally, the influence of friendships was highlighted by the interviewees, which has already been observed and corroborated by Weiss and Petlichkoff [39], Bustard et al. [40], and Lorenzo and Calleja [6]. On the other hand, all the women athletes stated that it was important to combine their studies with sports, although they emphasized the difficulty of doing so. In this sense, Morillas et al. [41] state that this may be because the women athletes have to face different problems such as the distribution of time for study and competition. They also referred to the fact that their studies increased their future job opportunities once they had finished their sporting career, which is in line with the findings of Álvarez et al. [42]. Additionally, the results of this study showed that for all the interviewees it was important to balance friendships with sport, but that it was not easy either. In this regard, Ferrer-Caja and Weiss [43] and Weiss and Smith [44] emphasize that young people’s sport commitment and long-lasting practice are closely related to the relationship with friends and peers, both inside and outside sport, as social relationships involve a great deal of social and emotional interaction [45]. Finally, and to conclude this dimension, some women athletes pointed out the low visibility of Paralympic sport in the media, and even less of women’s Paralympic sport, which is far from desirable and does not help [46]. For Dull-Tepper [47], radio and TV sports commentators and presenters do not provide the audience with sufficient information about Paralympic sport; therefore, there is no adequate coverage of Paralympic sport.

Regarding the psychological dimension, all the Paralympic women athletes considered the importance of psychological aspects to be of maximum significance, not only during the training process but also when they reached the elite level. Along these lines, Sánchez and León [48] state that, at present, these aspects play a relevant role in the performance of athletes. Concerning the relevance of receiving psychological support, most of the women athletes claimed that it is of vital importance. However, not all athletes worked with a psychologist, perhaps due to a lack of access or financial resources. Thus, psychological training is becoming increasingly relevant and decisive in high-performance sports [49], and not having access to a sport psychologist can even generate stress [50]. Furthermore, the results of this study showed that some of the women athletes who had worked with psychologists were satisfied with the performances obtained after the improvement in their behavior and the development of their potential [51]. Concerning competition, most of the women athletes interviewed believed that emotions and motivation played a fundamental role. In this line, Valdés [52] points out motivation as one of the most relevant aspects in the preparation and training of the athlete, as it functions as a regulator of energy and emotions which facilitates the achievement of objectives. Moreover, self-motivation in athletes is related to greater adherence to sports practice [53]. Similarly, all Paralympic women athletes reported self-confidence as another determining aspect in sport, especially in elite sport, as it allows for better sporting performance [54]. Finally, approximately half of the women athletes interviewed revealed having received pressure from coaches. Nevertheless, García et al. [55] claim that high pressures and stress trigger certain negative emotions and that anxiety alters the cognitive processes and behaviors of the athletes. Furthermore, Ramos [56] states that pressure can be a determining factor, as it often imposes extreme training levels on the athletes, which, together with the high pressures to which athletes are subjected in competitions, means that being psychologically strong can be key to obtaining sporting success [14,57].

According to all the Paralympic women athletes, both technique and tactics were important in their training as athletes, and they believed that these aspects should be worked on in the initial stages of sports training, which coincides with what was stated by Amador [58] and Ruíz and Arruza [59]. Regarding the physical fitness dimension, again, all the interviewees deemed physical fitness to be important in the process of sports training and performance, which has been emphasized by numerous authors [60,61]. As for the person in charge of their physical preparation, most of the interviewed women athletes stated that the main responsible people were the coaches and national selectors. On this point, Moya [62] states that it is important for this area to be led by a physical trainer, as he/she is the specialist. On the other hand, when the women athletes were asked about how they had worked on physical fitness, most of them claimed that they carried it out in an integrated way together with the technical–tactical aspects, while others trained their physical fitness independently or separately. In this way, it should be noted that some authors [63,64] stress the relevance of carrying out integrated training due to the numerous benefits it has at all levels: performance, physical fitness, technical–tactical and psychological, as long as the planning is correct. 

In terms of the perceived barriers and facilitators dimension, the women athletes interviewed reported that their sporting success was based on different aspects, ranging from training and discipline to self-confidence and motivation. In this line, Liones and Samalot [65] argue that some athletes did not succeed because, among other things, they did not receive the appropriate teaching, at the right time and by people who were prepared, trained, and qualified to do so, and because of poor or no psychological preparation. On the other hand, when the interviewees were asked about what their ideal sports training model would be like, most of them suggested that the training process should be more enjoyable and that the pressure to achieve results too early should be removed. Similarly, Shields et al. [66] highlighted in their study that if athletes thought more about having fun as well as competing, they would be more motivated and would forget about some of the difficulties. In terms of gender barriers, the vast majority of the interviewees stated that they had not felt discriminated against for this reason, even though hierarchical bases of gender ideology and the meaning of masculinity have been ingrained within sports [67] and that women’s participation in sport has often been criticized and rejected [68]. In terms of barriers due to a disability, half of the women athletes interviewed underlined the poor preparation of some sports facilities, sports equipment, etc. Other barriers mentioned by the interviewees were economic barriers. This aspect is significant because, according to Rodríguez [69], it is one of the reasons why some people with disabilities abandon the sport. Furthermore, the media are also involved in the discrimination suffered by Paralympic sports, especially women’s Paralympic sports. In this sense, Leardy [70] claims that, although journalists and society consider Paralympic sport to be becoming more and more widespread, it is a reality that non-Paralympic sport continues to occupy the front pages of the main sports newspapers.

To answer the question of the research presented at the beginning of the manuscript, it should be said that in the process of training Spanish Paralympic women, many factors are involved, among which the influence of the coach, the family, and psychological aspects such as motivation and self-confidence stand out. In addition, the main barriers encountered by athletes refer to economic aspects and the treatment they receive from the media.

## 5. Conclusions

The coach has a decisive influence on the training and performance process of Paralympic athletes. As for the close social environment, family and friends are also crucial during this process. Moreover, Paralympic women athletes find it difficult to combine academic education with high-performance sport and high-performance sport with social relationships. Regarding psychological aspects, athletes consider it necessary to work with specialists to control emotions and increase motivation and self-confidence, as well as to reduce stress and anxiety and manage pressure. Additionally, the women athletes believe it is appropriate that the technical, tactical, and physical fitness aspects are worked on in a joint or integrated manner. Finally, it should be noted that the training process and sporting performance of Paralympic women athletes are conditioned by several barriers, including economic, social, architectural, and disability barriers.

The strengths of the study lie in the fact that this research highlights the factors that can have the most influence on the process of formation of Paralympic athletes, highlighted by the athletes themselves. In this sense, regarding practical applications, the considerations of this study could be considered by the technical teams working with Paralympic athletes, as well as by the competent bodies, to improve the sports training process of these women athletes. In this way, sports federations and associations should make it easier for Paralympic women athletes to work with physical trainers and sports psychologists. In addition, the competent institutions must implement policies that increase the visibility and social recognition of these athletes, as well as remove barriers due to a disability and economic reasons. 

## Figures and Tables

**Figure 1 sports-11-00057-f001:**
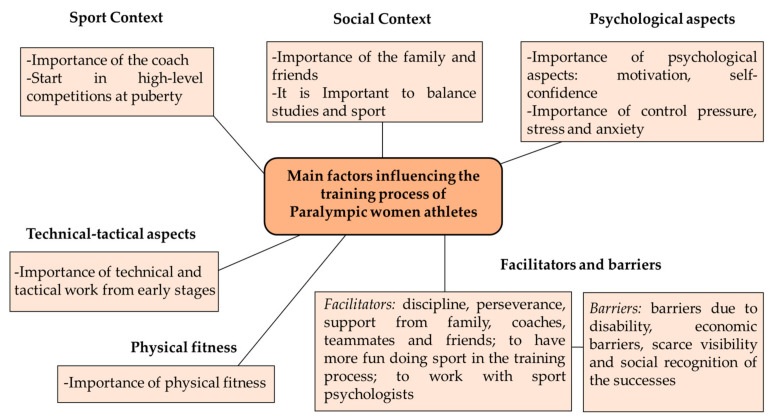
Main factors influencing the training process of Paralympic women athletes.

**Table 1 sports-11-00057-t001:** Categories, descriptions, and examples of the Sports Context dimension.

Categories	Description	Examples
Age	This is the age at the beginning of the practice	At 5 years
Memories	Comments on memories (positive or negative)	Emotionally, I remember it as very positive. Very cozy. I had good feelings
Other Sports Practiced	Everything related to the practice of sports other than the sport you practice	At school. Outside school, dancing and skating
Clubs and Sports schools	Comments on anything related to different clubs or schools of which you have been a member	The first club I was at started growing in water polo and losing in swimming. There were few spaces, times, and coaches for swimming, so I changed clubs
Reasons	These are the different reasons why you started playing sports or competing	Medical reasons mainly
Coach	Everything related to coaches and teachers that you have had throughout your sporting life	At some point, I felt pressured by the coaches to achieve results
Competition	Related aspects or comments about the competition or high competition	I was dedicated to high competition for 12 years
Performance	Comments related to sports performance	Just a few months before the Paralympic Games in Atlanta, my mother talked about the possibility of participating in the Games. It is the moment I open my mind and see that I can do more things at the sporting level

**Table 2 sports-11-00057-t002:** Categories, descriptions, and examples of the Social Context dimension.

Categories	Description	Examples
Family	Comments where you allude to your close environment (family)	My parents, because they have allowed me to practice it and have encouraged me always
Studies and Training	Everything relating to studies and/or academic training	Yes, I have early childhood education and psychopedagogy
Friends and colleagues	Everything about friends and/or teammates	My closest friends if they are related to sport
Other Persons	Comments about other people who are not family, friends, or teammates	Know how to manage what others tell you to
Means of Communication	Allusions to the media	I think they could watch more sports. The media should help and give us more impact. For example, swimming is seen very little in Olympic sport

**Table 3 sports-11-00057-t003:** Categories, descriptions, and examples of the Psychological Context dimension.

Categories	Description	Examples
Importance Aspects Psychological	The degree of importance the athlete gives to the psychological aspects	I think the psychological aspects have been very important
Aspects Highlights	Psychological aspects that have allowed they to reach the elite level	It helped me a lot to learn how to filter comments. Listening to some things and not listening to other advice that came from people that I thought were very important to me
Psychological Support	Comments on the importance of psychological support	Within my training as an athlete if I work with psychologists
Emotions	Comments on the emotions and feelings experienced by the athlete during her sporting life	With controllable nerves. Although in my good times, they have been one way and in the not so good they have been another
Motivation	Comments cited on the motivation that drives you to start or continue to practice sport	Very high. That is the high point. You have been trained for that
Self-confidence	Comments on the confidence you have in yourself to accomplish or achieve something	It has a lot of influence. When we went to the international championships beforehand, we did concentrations, and they gave me a lot of confidence in the training I had done. That is why in the championships that I participated in during the season, I was more unsure whether I was well prepared or not, but in the internationals, I went with a lot of certainties that I was well prepared
Pressure/stress/ Anxiety	Everything related to the pressure/stress you have suffered, or not, throughout your sporting life	Yes, it has happened to me many times. Additionally, goals that I have not set for myself, but have been imposed on me

**Table 4 sports-11-00057-t004:** Categories, descriptions, and examples of the Technical–Tactical Aspects dimension.

Categories	Description	Examples
Technical importance	Importance of technical aspects during the training process	The technical aspects are very important. They are the foundation of everything
Tactical importance	Importance of tactical aspects during the training process	They were also important. You have to know your strengths, if it is the exit, if it is the flip, if it is the swim, and you have to know where you have to make the most effort and where you have to reserve the energy. It is very important and at first, when you start, you are not able to handle it
Own characteristics	Opinion on your technique and tactics	My success as an athlete has depended on technique rather than tactics
Methodology	Comments and opinions on the different ways to train both technique and tactics	I think I have worked them properly, otherwise I would not have had those results

**Table 5 sports-11-00057-t005:** Categories, descriptions, and examples of the Physical Condition dimension.

Categories	Description	Examples
Importance of physical fitness	The degree of importance that physical fitness is considered to have during the training process	During my training, physical preparation has been very important. It is a physical sport
Work and recommendation on physical fitness	Opinion on whether you have worked well or not in physical fitness throughout your sporting life and different recommendations for working on it	I think a mix. Some sessions independently and when you already have a base, in an integrated way. We trained to a greater extent in an integrated way
Important Physical Qualities	Opinion on the most important physical qualities to reach the elite level	Endurance is the most important physical ability to reach the elite
Own Physical Qualities	Opinion you have about your physical qualities	The qualities that prevailed with respect to the other athletes were stamina and technique

**Table 6 sports-11-00057-t006:** Categories, descriptions, and examples of the Barriers and Facilitators dimension.

Categories	Description	Examples
Success	Comments or opinions on what have been the causes of sporting success	It is not something concrete, but a combination of circumstances and sometimes also coincidences because it was by chance that my parents introduced me to swimming and then also the work that I have been doing with coaches who have supported me
Failure	Comments or opinions on what are the reasons that cause failure in some athletes	They aim high very soon. Self-confidence is important, but we have to be realistic. You have to go setting achievable goals
Facilitators	Comments or views on how the barriers they face could be removed, reduced or facilitated	I was the only one with physically disabled people in the village where he trained. I was very close to where I lived and, in addition, in the training, I did not have to adapt myself to people with physical disabilities, but I had my training
Proposal Formation Model	Opinion on what is the ideal training model for Paralympic athletes and on the changes it would make to their training process	Growth as a person, integral development, not just sports growth, must be taken into account
Barriers by gender	Comments on the different difficulties that the athlete has encountered because she is a woman	As a woman, I do not think I had any barriers, but I do because of my disability
Barriers by disability	Comments on the different difficulties that the athlete has encountered because of having a disability	For example, we have not had the same financial support as non-disabled sportsmen
Other barriers	Other challenges	I put them together until a moment came when I had reached the highest point in the world of sports. I have lived it before, and I did not want to live downhill. Studies were above sports. If it were like footballers who make money, you might think so, but it was not the case.

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
