# Peer review of "Factors Influencing the Training Process of Paralympic Women Athletes"

_sports, 2023, doi:10.3390/sports11030057_

Round 1

Reviewer 1 Report

- consider referring to women athletes rather than sportswomen.

- consider referring to athletes rather than sportspersons.

- Overall, the introduction section failed to present a convincing rationale as to why this study is needed, and what the benefits will be both to science and society, especially for sport-related professionals/researchers. Without this, it comes across as more of a limited research note.

-        It was not clear to me what was being examined and the extent to which results could contribute to the literature, but if the core aims are better explained, this may become clearer. I suggest breaking down the introduction section into subheadings, helping the authors and readers to understand the sequence flow of your research.

- English writing needs substantial revision as some sentences are unclear (e.g., lines 46-49) or hard to read (e.g., lines 58-61).

- The authors do not present limitations of existing studies.

- Concepts of psychological variables should be described for readers.

- a Figure would be welcomed considering the results of the interviews. A network analysis would increase the importance and contribution of this study.

- practical implications should be explored more thoroughly.

Author Response

COVER LETTER

Manuscript ID: sports-2224498. Type of manuscript: Article. Title: Factors influencing the training process of Paralympic women athletes (modified title).

Reviewer 1’s comments and suggestions for authors

Details of the revisions and responses

- Consider referring to women athletes rather than sportswomen

This suggestion has been taken into account throughout the text

- Consider referring to athletes rather than sportspersons

This suggestion has been taken into account throughout the text

- Overall, the introduction section failed to present a convincing rationale as to why this study is needed, and what the benefits will be both to science and society, especially for sport-related professionals/researchers. Without this, it comes across as more of a limited research note

The need for the study and its benefits have been convincingly justified, both for professionals and researchers related to sports, and from a social point of view

The knowledge, study and research of all the factors that influence the training pro-cess of women athletes can help coaches, physical trainers, psychologists and other professionals involved in sports training to know how and to what extent these factors affect the training process of athletes, with a view to improving the training process of athletes, making it more complete and rigorous, and optimizing their performance, thus increasing their chances of achieving better sporting results [8]. It is well known that the planning of the training process is essential, as it helps athletes to acquire a good foundation, which is fundamental to compete at the elite level [16]. In addition, from a social justice point of view, conducting this study is justified because it can contribute to increasing the visibility of women athletes with disabilities, not only among researchers, academics, and sports training professionals, but also in the rest of society, promoting awareness of the importance of the role of these women athletes

- It was not clear to me what was being examined and the extent to which results could contribute to the literature, but if the core aims are better explained, this may become clearer. I suggest breaking down the introduction section into subheadings, helping the authors and readers to understand the sequence flow of your research

The aim of the research carried out have been better explained

The aim of this research was to analyse the factors influencing the sport training process of Spanish Paralympic women athletes (social, sporting, psychological, technical-tactical factors, physical condition, as well as barriers and facilitators) who had won at least one medal (gold, silver or bronze) in the 21st Century Paralympic Games (from Sydney 2000 to Tokyo 2020)

The introductory section has been divided into subsections in order to help understand the sequential flow of the study carried out:

1.1.  The process of sports training

1.2.  Factors influencing the training process of Paralympic women athletes

1.3.  Need for research

- English writing needs substantial revision as some sentences are unclear (e.g., lines 46-49) or hard to read (e.g., lines 58-61)

English writing has been revised

- The authors do not present limitations of existing studies

Limitations of the study appear at the end of the discussion section

As for the limitations of this study, the main one is that the existing literature on this subject is quite scarce, especially in relation to Paralympic women. Another limitation was that it was not possible to interview some women athletes due to their busy schedules

- Concepts of psychological variables should be described for readers

The concepts of psychological variables have been described

emotions, motivation, self-confidence, pressure, stress and anxiety. In this sense, emotions are understood as a complex reaction pattern, which involves experiential, behavioural and physiological elements; motivation is defined as the impulse that gives purpose to behaviour; self-confidence refers to confidence in one's own abilities and abilities; pressure is understood as excessive or stressful demands that influence the way one thinks, feels or acts; stress is the physiological or psychological response to internal or external distressing factors; while anxiety is characterized by the learning of emotions and behaviours. and somatic symptoms of tension that a person may manifest

- a Figure would be welcomed considering the results of the interviews. A network analysis would increase the importance and contribution of this study

A Figure summarizing the main results of the study carried out has been introduced

- practical implications should be explored more thoroughly

the practical implications of the research carried out have been broadened

In this sense, regarding practical applications, the considerations of this study could be considered by the technical teams working with Paralympic athletes, as well as by the competent bodies, in order to improve the sport training process of these women athletes. In this way, sports federations and associations should make it easier for Paralympic women athletes to work with physical trainers and sports psychologists. In addition, the competent institutions must implement policies that increase the visibility and social recognition of these athletes, as well as remove barriers due to disability and economic reasons

Reviewer 2 Report

Dear Authors,

Congratulations for the work developed so far. This paper is very pertinent and useful for sports literature. I think the paper could be accepted for publication, but currently there are some issues that need to be solved. I hope the comments provided can help you improving the paper.

 All the best

Author Response

COVER LETTER

Manuscript ID: sports-2224498. Type of manuscript: Article. Title: Factors influencing the training process of Paralympic women athletes (modified title).

Reviewer 2’s comments and suggestions for authors

Details of the revisions and responses

1 – Please, remove or reword to avoid repetition. You expressed the same idea few sentences above

Taking into account the reviewer's suggestion, to avoid confusion and to guide the reader, two paragraphs have been introduced to clarify the content of the introduction. On the one hand, it discusses the training process of athletes in general and, on the other hand, the factors that may influence the training process of Paralympic athletes

2 – “competences bodies”? Please, clarify

The term "competent bodies" has been clarified in the discussion section

3- Sorry but the meaning of this sentence/statement is hard to follow. This sounds to me not a feature of sportswomen, but from sport in a general fashion.

Reworded the sentence suggested by the reviewer

4 - This first paragraph is very descriptive. Moreover, there is not a clear and easy link among sentences. Please, revise it accordingly

The reviewer's suggestion has been taken into account

The training process of the women athlete goes through a series of stages, parallel to its evolutionary development [1]. This process is influenced by a wide variety of psycho-logical, social, and biological factors [2]. In this sense, Giménez [3] affirms that the athlete will go through a series of stages in which she will learn and be trained in different con-tents (physical, technical, tactical and psychological) adapted to her biological and psy-chological characteristics. In this way, Cárdenas and López [4] consider that sports train-ing is influenced by methodological, psychological, genetic and physiological factors, which significantly affect the training of athletes. In this sense, also Castejón et al. [5] con-sider that, in addition, the motor, cognitive and affective aspects of each athlete should be taken into account. In this sense, Lorenzo and Calleja [6] suggest analyzing the process of training high-performance athletes and the factors that influence it, in order to find the op-timal conditions for the development of competencies that can be applied to the process of training new talented athletes

5 – General appointment for the introduction: please, try to demonstrate and explain in a more critical way, what is the pertinence of this study. Also, add at this section the relevance of using a qualitative approach

The introduction has broadened the justification and relevance of the study carried out, as well as the use of a qualitative approach

The knowledge, study and research of all the factors that influence the training process of women athletes can help coaches, physical trainers, psychologists and other pro-fessionals involved in sports training to know how and to what extent these factors affect the training process of athletes, with a view to improving the training process of athletes, making it more complete and rigorous, and optimizing their performance, thus increasing their chances of achieving better sporting results [8]. It is well known that the planning of the training process is essential, as it helps athletes to acquire a good foundation, which is fundamental to compete at the elite level [16]. In addition, from a social justice point of view, conducting this study is justified because it can contribute to increasing the visibility of women athletes with disabilities, not only among researchers, academics, and sports training professionals, but also in the rest of society, promoting awareness of the importance of the role of these women athletes. In this sense, the qualitative approach was selected in order to examine the way in which the subjects perceived and experience the phenomena that surrounded them, deepening in their views, interpretations and meanings, and because the subject of the study has been little explored [17]

6 – Are these values about age?

Yes. Most of the women athletes were retired and many of them, at the time of the interviews, were over 40 years old

7 - Please, clarify how and why qualitive research aligns with study's purpose

This question has been dealt with in the introductory section

8 – why did you use a pre-stablished interview? Did you ask more questions than the ‘questionnaire’? These reasons have to be mentioned

The reasons for using the semi-structured interview of Robles et al. (2016) have been described and it is explained why a dimension was added

This tool was used because the researchers knew it well and hoped, as in the past, to obtain robust results after its application. With the idea of completing this tool, the dimension "facilitators and barriers" was added, due to the importance of the subject for athletes with disabilities and due to its scarce study in the specialized literature

9 – Please, mention also the advantages and disadvantages of using a structured interview. I suggest linking it with the features of the sample interviewed.

The advantages and disadvantages of using structured interview have been added

In this regard, it should be noted that some of the advantages of using an interview are that the interviewees can provide historical information and, in addition, it allows some control of the interviewer when it comes to including and addressing the issues and questions. Regarding the limitations of the use of interviews, the fact that the data is “filtered” by the views of the participants and that not all subjects have the same abilities to express themselves verbally

10 – Please, avoid the repletion on the meaning of each one od the categories at the beginning of a new subsection. You have already mention it above, at the methods section

As suggested by the reviewer, the explanation of each dimension has been removed from the results section because they have already been explained in the method section

11 – I think the authors did a great work on both sections. However, I strongly recommend adding/extending the practical applications of this study

As suggested by the reviewer, the practical applications of the study have been broadened

In this sense, regarding practical applications, the considerations of this study could be considered by the technical teams working with Paralympic athletes, as well as by the competent bodies, in order to improve the sport training process of these women athletes. In this way, sports federations and associations should make it easier for Paralympic women athletes to work with physical trainers and sports psychologists. In addition, the competent institutions must implement policies that increase the visibility and social recognition of these athletes, as well as remove barriers due to disability and economic reasons

Reviewer 3 Report

I tried to give my best and hope my comments will be helpful to the authors. Also, it is necessary to show results in tables or figures. I gave an explanation. Read the manuscript carefully. Probably I missed some words when I tried to correct all words in American English.

Recommending for publication because of a very important topic!

File attached.  

Author Response

COVER LETTER

Manuscript ID: sports-2224498. Type of manuscript: Article. Title: Factors influencing the training process of Paralympic women athletes (modified title).

Reviewer 3’s comments and suggestions for authors

Details of the revisions and responses

General comment for this part is that

it is necessary to have tables or better: figures where the

results are shown. Please take into account that your article in

future has to be cited and that researchers don’t have days and

weeks to read it all carefully. My opinion is that it is always

better to have data in tables and/or figures, as well

A Figure summarizing the main results of the study carried out has been introduced

All suggested changes related to English language have been made

Round 2

Reviewer 1 Report

Thank you for revising the manuscript according to the comments provided by the reviewers.

Author Response

Thank you very much for your invaluable help.

Reviewer 2 Report

Dear Authors,

I hope you are doing very well.

I thank for the work developed so far. Unfortunatly, some important issues remain unresolved. Thereby, I recommended major revisions once again. Please, considerer the comments below.

Kind regards,

1 - the english writing needs to be carefully revised througout the paper. Also, shorter sentences and explicit messages are needed. Otherwise, in my opinion the paper will be unsuitable for publication. 

2 (lines 32-46) - This first paragraph is very descriptive. Also the wirting is very confuse... You start almost all sentence with similar expressions "in this sense", "in this way"...; Moreover, what is the relationship of this information with your study? You have to create explicit links. Please, revise it accordingly

3 (line 51) - Please, remove 'an'

4 (lines 63 - 66) - There is a large repetition on this sentence. Please, reword it

5 (lines 69-74) - This sentence is too long. The information is hard to follow. Please, revise it.

6 (lines 76-79) - Sorry but I do not agree. The pertinence of your work is not related with social justice, but with a lack of investigations within this scope. If there is not scientific information, the practice cannot be improved... its not only about justice, its mainly about being useful...

7 (lines 80-83) - Again, these arguments are not strong enough... 

8 (lines 108-109) - "and due to its scarce study in the specialized literature".  These kind of information must be mentioned at the introduction section

9 (line 133) - Again, there is repeted information "and abilities"

10 (lines 694-698) - I suggest removing. These are not limitations, but facts.

Author Response

Sports 2224498. Factors influencing the training process of Paralympic women athletes

1 - the english writing needs to be carefully revised througout the paper. Also, shorter sentences and explicit messages are needed.

The text has been revised.

2 (lines 32-46) - This first paragraph is very descriptive. Also the wirting is very confuse... You start almost all sentence with similar expressions "in this sense", "in this way"...; Moreover, what is the relationship of this information with your study? You have to create explicit links. Please, revise it accordingly

The text has been revised.

3 (line 51) - Please, remove 'an'

Ok

4 (lines 63 - 66) - There is a large repetition on this sentence. Please, reword it

This sentence has been removed as suggested by the reviewer.

5 (lines 69-74) - This sentence is too long. The information is hard to follow. Please, revise it.

The sentence has been modified in line with the reviewer's suggestions.

6 (lines 76-79) - Sorry but I do not agree. The pertinence of your work is not related with social justice, but with a lack of investigations within this scope. If there is not scientific information, the practice cannot be improved... its not only about justice, its mainly about being useful...

The sentence has been modified in line with the reviewer's suggestions.

7 (lines 80-83) - Again, these arguments are not strong enough... 

The arguments have been improved.

Studies on the triangle gender, disability and sport have little presence in the specialized literature [16] despite the great growth of the Paralympic Games and the incorpora-tion of women in this event [17]. Athletes with disabilities have to overcome structural, so-cial, and economic barriers. In addition, Willis et al. [18] highlights the existence of a very limited understanding of the mechanisms and processes of training disabled athletes that allow them to participate in competitive-oriented physical activities. Knowledge of all these aspects can help coaches, psychologists and other professionals to improve the sports training of these athletes [8, 19]. The scarcity of research focused on the analysis of these factors, and in this context, justifies this work. Also, this study is justified because it can contribute to increasing the visibility of women athletes with disabilities, not only among researchers, academics, and sports training professionals, but also in the rest of society, promoting awareness of the importance of the role of these women athletes. The qualitative approach was selected to examine the way in which the subjects perceived and experienced the phenomena that surrounded them, deepening in their views, interpreta-tions and meanings, and because the subject of the study has been little explored [20].

8 (lines 108-109) - "and due to its scarce study in the specialized literature". These kind of information must be mentioned at the introduction section

Ok

9 (line 133) - Again, there is repeted information "and abilities"

The sentence has been modified in line with the reviewer's suggestions.

10 (lines 694-698) - I suggest removing. These are not limitations, but facts.

This sentence has been removed as suggested by the reviewer.
